# A Siderophore Analog of Fimsbactin from *Acinetobacter* Hinders Growth of the Phytopathogen *Pseudomonas syringae* and Induces Systemic Priming of Immunity in *Arabidopsis thaliana*

**DOI:** 10.3390/pathogens9100806

**Published:** 2020-09-30

**Authors:** Fabrice Betoudji, Taha Abd El Rahman, Marvin J. Miller, Manuka Ghosh, Mario Jacques, Kamal Bouarab, François Malouin

**Affiliations:** 1Département de Biologie, Faculté des Sciences, Université de Sherbrooke, QC J1K 2R1, Canada; fabrice.betoudji@usherbrooke.ca (F.B.); taha.abd.el.rahman@usherbrooke.ca (T.A.E.R.); 2Department of Chemistry and Biochemistry, University of Notre Dame, Notre Dame, IN 46556, USA; mmiller1@nd.edu; 3Rosetree Corporate Center, Hsiri Therapeutics, Inc., Media, PA 19063, USA; mghosh@hsiritherapeutics.com; 4Département de Pathologie et Microbiologie, Faculté de Médecine Vétérinaire, Université de Montréal, Saint-Hyacinthe, QC J2S 2M2, Canada; mario.jacques@umontreal.ca

**Keywords:** iron, siderophores, *Acinetobacter*, *Arabidopsis thaliana*, *Pseudomonas syringae*, systemic resistance, PGPR

## Abstract

Siderophores produced in soil by plant growth-promoting rhizobacteria (PGPRs) play several roles, including nutrient mobilizers and can be useful as plants defense elicitors. We investigated the role of a synthetic mixed ligand bis-catechol-mono-hydroxamate siderophore (SID) that mimics the chemical structure of a natural siderophore, fimsbactin, produced by *Acinetobacter* spp. in the resistance against the phytopathogen *Pseudomonas syringae*
*pv tomato* DC3000 (*Pst* DC3000), in *Arabidopsis thaliana.* We first tested the antibacterial activity of SID against *Pst* DC3000 *in vitro*. After confirming that SID had antibacterial activity against *Pst* DC3000, we tested whether the observed *in vitro* activity could translate into resistance of *Arabidopsis* to *Pst* DC3000, using bacterial loads as endpoints in a plant infection model. Furthermore, using quantitative polymerase chain reaction, we explored the molecular actors involved in the resistance of *Arabidopsis* induced by SID. Finally, to assure that SID would not interfere with PGPRs, we tested *in vitro* the influence of SID on the growth of a reference PGPR, *Bacillus subtilis*. We report here that SID is an antibacterial agent as well as an inducer of systemic priming of resistance in *A. thaliana* against *Pst* DC3000, and that SID can, at the same time, promote growth of a PGPR.

## 1. Introduction 

With the exception of a few organisms, such as lactobacilli and Borrelia burgdorferi [1,2], virtually all living organisms, including plants, require iron for biological functions. In plants, iron is required for many physiological functions, such as photosynthesis, chlorophyll biosynthesis, and respiration. The soil is the main source of iron acquisition for plants that assimilate it from their roots [3]. Although the third most abundant element in the earth’s crust [4], iron is paradoxically in limited quantity for plants because its availability depends on many factors, such as the pH of the soil and the redox potential [5]. Under aerobic conditions and at higher pH, iron is mainly present in its less soluble ferric oxidized form (Fe^3+^) and is therefore not available for uptake by plants. 

Because iron is vital to both plants and pathogens, its acquisition is one of the most important adaptative responses. Like mammals, plants secrete a variety of proteins and ligands to prevent invading pathogenic bacteria from accessing iron [6]. To counter this iron restriction by plants, pathogenic bacteria produce siderophores which have higher affinity for iron than phytosiderophores (PS) or other plant iron carriers, to acquire iron from their host, and promote infection [7]. Siderophores are important microbial virulence factors [8,9,10]. Mammals respond to the presence of bacterial siderophores by secreting lipocalin 2 [11], which confiscates the iron-loaded siderophore and prevents the complex from returning to the producing bacteria. Although their role in iron homeostasis is yet to be elucidated, lipocalins are also found in plants [12]. The plant-pathogen competition for iron has an immediate effect on plant immunity; for example, siderophores produced by *Dickeya dadantii* induce the expression of ferritin-1 (FER1), an iron storage gene, in *Arabidopsis* [9]. Likewise, the beneficial plant growth-promoting rhizobacteria or PGPR, positively affect plant resistance through iron, by the induced systemic resistance (ISR) mechanism [13,14]. Induced systemic resistance is described as an increased protective ability acquired by plants against a wide range of pathogens following stimulation by beneficial bacteria. In the ISR mechanism, certain bacterial components such as flagellin or lipopolysaccharide (LPS) termed PAMPs or MAMPs (pathogen-or microbe-associated molecular patterns) are recognized by receptors expressed by plants called PRRs (Pattern-Recognition Receptors); and an immune defense response is initiated upon recognition of those PAMPs or MAMPs [15]. Certain bacterial components, such as LPS and pseudobactin, were found to trigger the ISR in different plant species when applied to roots and leaves prior to challenges by plant pathogenic agents [16]. This first line of defense in plants is called PTI or PAMP-Triggered Immunity, and can stop most pathogens without harming the plant [17]. However, some pathogens are able to interfere with the PTI defense by producing proteins called effectors, which interact directly with PRRs and thus suppress their activity. To respond to the presence of effectors, plants possess a second line of defense which involves expression of a specific protein that targets and inhibits the effector. This second line of plant defense is called ETI, for Effector-Triggered Immunity, and is highly effective and specific [17]. Implementation of PTI or ETI defenses in plants involves the recognition of pathogens and synthesis of signaling defense molecules, including hormones. The most characterized of these hormones are salicylic acid (SA), ethylene (ET), and jasmonic acid (JA) [18]. These low molecular weight compounds are powerful inducers of a number of defense genes that trigger most of the responses required to neutralize pathogens. 

It has recently been reported that *Acinetobacter* can act as a PGPR [19]. The genus *Acinetobacter* is also known to produce the siderophore fimsbactin [20,21]. Since ISR activated by PGPR against phytopathogens can be triggered by siderophore-mediated competition [16], we set out to assess the potential protective effects of a synthetic siderophore, a mixed ligand bis-catechol-mono-hydroxamate analog (SID in Figure 1), acting as a mimetic of the natural *Acinetobacter* siderophore, [22]. The *in vitro* effect of several natural and synthetic siderophore analogs including synthetic fimsbactin analog SID, against the plant phytopathogen *Pseudomonas syringae pv tomato* DC3000 (hereafter referred to as *Pst* DC3000) and its effect on resistance to the same pathogen in *Arabidopsis thaliana* was therefore evaluated in this study. We also assessed the impact of this synthetic siderophore on the growth of PGPR *Bacillus subtilis* and *Acinetobacter baumannii*. A flowchart summarizing the steps taken in this study is shown in Figure 2.

## 2. Results

### 2.1. SID Inhibits Pst DC3000 but not B. subtilis or A. baumannii Growth In Vitro 

The iron acquisition machinery in bacteria is overexpressed in iron-depleted environments [25]. The growth of bacterial pathogens may be significantly compromised if the iron acquisition machinery they express is not adequate for the environment they face. To test whether *Pst* DC3000 possesses iron transport systems for the uptake of the synthetic siderophores SID and CAT (Figure 1B,D), we depleted LB broth by treating it with the chemical iron chelator, 2, 2’-bipyridyl, and supplementing it with CaCl_2_, in order to minimize possible chelation of calcium. In this iron-depleted environment, *Pst* DC3000 is expected to upregulate its iron acquisition systems; and if these include adequate and specific siderophore uptake systems, bacteria should be able to actively grow compared to what is observed in the iron-depleted medium where no siderophore was provided. If *Pst* DC3000 does not have the uptake systems for the provided siderophores or if its own siderophores cannot be more efficient than the added siderophores, the additive effect of the iron-chelating siderophores to that of the bipyridyl would exacerbate the lack of iron in the medium, and further limit bacterial growth (antibacterial activity). First, Figure 3A shows that there is no growth promotion or inhibition of *Pst* DC3000 by the hydroxamate siderophore DFO (up to 32 µM), as previously reported by others [26] and reveals that *Pst* DC3000’s own siderophores (e.g., pyoverdine) can efficiently compete and provide iron to bacteria in this iron-restricted environment. On the other hand, Figure 3A shows that the synthetic SID inhibits *Pst* DC3000 growth, measured by OD_595 nm,_ after 24 h of incubation, at all concentrations tested (≥2 µM) compared to the control without any siderophore added. The bis-catechol CAT also inhibits *Pst* DC3000 growth although it requires higher concentrations (≥16 µM). This shows that SID and CAT are efficient iron chelators and that they cannot be used by *Pst* DC3000 for growth. Since the synthetic SID has a better *in vitro* inhibitory effect against *Pst* DC3000 than CAT, SID was selected for further testing. In addition, for SID to have applications in agriculture, it was important to establish that it did not interfere with plant beneficial bacteria. To test that hypothesis, we studied *B. subtilis* as a representative PGPR. We also used *A. baumannii* to show that it does indeed possess SID uptake machinery to efficiently thrive in an iron-depleted environment. Thus, contrary to the inhibitory activity of SID against *Pst* DC3000 (Figure 3A), Figure 3B,C show that the mixed ligand SID promotes the growth of *B. subtilis* and *A. baumannii*, respectively. Since the mixed ligand SID is an analog of the natural mixed ligand (bis-catechol, mono-hydroxamate) siderophore, fimsbactin, produced by some strains of *Acinetobacter,* and that all *Acinetobacter* strains have the uptake machinery for structurally related siderophores [21], it is not surprising that the growth promotion of *A. baumannii* by the mixed SID is more significant than that observed for *B. subtilis*. Figure 3B shows; however, that *B. subtilis* can to some extent utilize this xenosiderophore under iron-depleted condition. This indicates that bacterial genera often considered PGPR, *i.e*., *Acinetobacter* and *Bacillus*, have the appropriate machinery for the utilization of the mixed ligand synthetic SID. The observed properties of SID (inhibitory activity on the pathogen *Pst* DC3000 and growth promoting activity for PGPR) make it suitable for use as a disease-preventing agent in plants. Subsequent experiments were aimed at determining the protecting effect of SID in plants against *Pst* DC3000. 

### 2.2. SID Promotes Priming of Arabidopsis Systemic Resistance against Pst DC3000

Our results showed that the mixed SID has antimicrobial activity against *Pst* DC3000 (Figure 3A), and interestingly it promotes the growth of PGPR (Figure 3B,C). Siderophores from beneficial bacteria can also activate plant immunity and induce local resistance against bacterial pathogens such as *Pst* DC3000 or pathogenic fungi such as *Botrytis cinerea* in *A. thaliana*, even when they are exclusively leaf infiltrated, and not root treated [6,26,27]. We thus tested whether the mixed ligand SID, in addition to its direct antimicrobial effect on *Pst* DC3000, was able to induce the plant immune response, leading to the activation of resistance against *Pst* DC3000. Since the synthetic SID imitates the chemical structure of the natural siderophore, fimsbactin, produced by the genus *Acinetobacter* and that PGPR are present in the rhizosphere and act through plant roots [28], it is then reasonable to think that if there was any effect of SID on plant immunity, this might occur through the plant roots. To test this hypothesis in plants, we poured SID (a 10-mL solution at 100 µM) directly in the pots, underneath the plant’s aerial parts, to assure it was in contact with the roots without touching the other parts of the plant such as the leaves. We then challenged the plant leaves with *Pst* DC3000, 24 h later. Bacterial counts in leaves were assessed on the day of the infection (Day 0) and 3 days after the infection (Day 3). As shown in Figure 4, SID induced resistance against *Pst* DC3000 in *Arabidopsis* compared to the control plants which roots were only pre-treated with the diluent used for SID (1% DMSO). Indeed, a statistically significant (*p* < 0.0001) 7-fold reduction in mean CFU counts of *Pst* DC3000 per mg of leaves was observed on Day 3 (3 days after infection). Note that no effect of SID was observed at lower concentrations (1 to 16 µM) and that solubility issues prevented us to test higher concentrations.

This indicates that the contact between SID and the roots of the plants induces some protection in *Arabidopsis* against *Pst* DC3000 at the leaf level. The next series of experiments thus aimed at the identification of the molecular actors involved in such a resistance. 

### 2.3. SA Pathway is Primed by SID 

To investigate the molecular actors implicated in the resistance conferred by the synthetic SID to *Arabidopsis*, we pre-treated the roots of two groups of plants with either SID (100 µM) or its diluent (1% DMSO); then leaves were challenged either with *Pst* DC3000 or a mock infection (*i.e*., the diluent used for bacteria, i.e., 10 mM MgCl_2_) as described in materials and methods. Challenged leaves were afterwards collected 8 h post-infection for RNA extraction and we quantified plant immunity markers, such as *PR1* for the SA pathway or for example *PDF1.2* for the JA pathway, since the SA and JA pathways are the two main essential and the most characterized pathways involved in the immune response against pathogenic agents [29]. Specific primers for marker genes of those pathways (Table 1) were used for quantitative polymerase chain reaction (qPCR). Figure 5A shows that the level of expression of *PR1* was upregulated in SID pre-treated group and challenged with *Pst* DC3000 compared to other treatment groups. This corresponds to a priming defense response pattern and indicates that the priming signal was sent from the roots since leaves were not in contact with SID. Contrary to the *PR1, PDF1.2* (Figure 5B) expression level was not primed by SID. These results suggest that the observed resistance of *Arabidopsis* to *Pst* DC3000 induced by SID is positively correlated with the priming of the SA pathway.

## 3. Discussion

To prevent and control some biotic enemies in agriculture, pesticides are widely used. However, their use is not without harm to the environment, to animals and to humans [30]. To reduce the use of pesticides, efforts should be made to better understand plants natural defense mechanisms, including interactions between plants and PGPR. PGPR help plants by mobilizing crucial but not readily available nutrients such as iron [31]. Moreover, PGPR facilitate plant growth directly by providing synthesized compounds such as phytohormones or by facilitating nutrients uptake (e.g., iron) and indirectly by promoting resistance against pathogens [32]. Among compounds produced by PGPR, siderophores provide nutrient competitive advantages to the producing microorganisms, and can protect plants from invading pathogens [31]. Moreover, microbial siderophores can act as activators of plant immunity [26]. However, this activation was until now thought to be local. Our results show that plant immunity elicited by siderophores can be systemic and induced from roots. These results have a biological significance, since PGPR are present in the rhizosphere and are able to induce systemic resistance. Moreover, our results show that the synthetic SID, analog of the natural siderophore fimsbactin, has a direct, antibacterial activity against the phytopathogen *Pst* DC3000 as shown in Figure 3A. This result indicates that this pathogen does not have iron uptake systems for SID and consequently SID induces iron starvation in the iron-depleted medium, an environment similar to that it would encounter in nature. Inversely, in similar *in vitro* growth conditions, strain members of the PGPR, namely *Acinetobacter* and *Bacillus*, grew better in the presence of SID (Figure 3C,D) compared to their growth controls without SID. The growth promotion was more pronounced for the *Acinetobacter* strain, which is known to use the natural siderophore fimsbactin. *B. subtilis* also benefits from SID, although to a lesser degree. Indeed, *B. subtilis* is known to possess iron uptake systems for exogenous siderophores [33]. 

Not only did the synthetic SID interfere with *Pst* DC3000 growth *in vitro*, but it also enabled the whole plant to resist the pathogenic bacteria *Pst* DC3000. More interestingly, the activity was observed after pre-treating plants at the roots with SID and challenging them with the pathogen at the leaf level 24 h later (Figure 4). Similar observations were made by other research groups before, although at the local level, where leaves were pretreated by purified natural siderophores (*e.g*., chrysobactin) or whole PGPR, and then infected with pathogenic agents [26,34,35]. Instead, in our experiments, roots were pretreated with SID and leaves infected with *Pst* DC3000. Paradoxically, it was observed that siderophores produced by pathogenic bacteria were involved in the promotion of systemic colonization; such was the case with chrysobactin from *Dickeya dadantii* [27]. However, it has also been noted that chrysobactin induces up-regulation of *AtFER1*, an iron storage gene from *Arabidopsis* involved in the protection against the phytopathogen *D. dadantii* [9]. This suggests that siderophores may play various roles depending on the conditions under which the plants find themselves. 

As for the molecular mechanisms of resistance induced by SID against *Pst* DC3000, our results show that the observed resistance is positively correlated with the systemic priming of the SA signaling pathway based on the expression of its marker gene *PR1* (Figure 5A). Similar *PR1* expression patterns in response to iron deficiency were reported by many other research groups but at local levels [9,26,27,36,37]. Conversely, there is no significant difference in the expression of the JA marker, *PDF1.2* (Figure 5B). 

To our knowledge, this is the first demonstration that a synthetic siderophore, that is an analog to a naturally produced bacterial siderophore, can be used to promote the plant immune defenses from the roots to the leaves. Based on the results presented in the present work, it would be conceivable to imagine the use of synthetic siderophores as biocontrol agents against phytopathogens in agriculture. Furthermore, it would be of great interest to investigate other types of synthetic siderophores (with various chemical functional groups) in agriculture to uncover whether they would have similar relative effects in plants, and to also determine the spectrum of activity of those synthetic siderophores against various pathogenic microorganisms including bacteria and fungi.

## 4. Materials and Methods

### 4.1. Synthetic Siderophores

The synthetic siderophores, i.e., the fimsbactin analog (SID) [22], and the bis-catechol (CAT) [24] were generously provided by Hsiri Therapeutics (Media, PA). The chemical structures of fimsbactin and myxochelin A [23], and those of their synthetic analogs SID and CAT, respectively, are presented in Figure 1. Deferoxamine (DFO) (Sigma-Aldrich, Oakville, ON, Canada) was used as a comparator in growth inhibition/promotion experiments (also shown in Figure 1).

### 4.2. In Vitro Growth Inhibition of Pseudomonas Syringae Pv Tomato DC3000 (Pst DC3000) by Synthetic Siderophores

*Pst* DC3000 was streaked from frozen stocks onto the King’s B (KB) selective agar medium containing rifampicin and kanamycin and incubated at 28 °C, 48 h prior to any experiment. To assess the growth inhibition potential of SID or CAT on the phytopathogen *Pst* DC3000 [38], lysogeny broth (LB) treated with an iron chelator, 2, 2’-bipyridyl (BiP) (Sigma-Aldrich, Oakville, ON, Canada) at a final concentration of 300 µM, and supplemented with CaCl_2_ (50 µg/mL), was used as the iron depleted broth. Twenty-four h leading up to the experiment, one colony of *Pst* DC3000 was picked from the KB agar plate and inoculated into a flask containing LB broth with the previously mentioned antibiotics and incubated with agitation (200 rpm) at 28 °C. On the day of the experiment, the synthetic siderophore SID or CAT was serially diluted in the iron depleted LB without antibiotic in a 96-well plate. *Pst* DC3000 was inoculated at a density of 10^5^–10^6^ colony-forming unit per mL (CFU/mL). The optical density (OD_595 nm_) of the plate was read 24 h after incubation at 28 °C, with shaking. Controls included non-supplemented iron depleted LB broth as well as broth supplemented with the natural siderophore deferoxamine DFO.

To check whether the synthetic SID could affect or be utilized by a reference PGPR, *Bacillus subtilis* NCIB3610 was used in the same assay described here for *Pst* DC3000.

### 4.3. SID-induced Resistance of Arabidopsis against Pst DC3000

To evaluate the plant systemic priming of immunity induced by SID, we used five-week-old *Arabidopsis thaliana*, ecotype Columbia-0 (Col-0) [38], grown in a 9 pot set (50 mL capacity per pot) in a growth chamber set at 22 °C –23 °C, at day time and 19 °C at night, with 70% relative humidity and a 12 h-photoperiod.

The synthetic SID, diluted in 10 mL of water to a 100 µM final concentration, was poured directly into the pot, underneath the plant, to maximize absorption by the roots. The mock was the solvent dimethyl sulfoxide (DMSO) to a final concentration of 1% in 10 mL, since SID was dissolved in that solvent. 

An overnight culture of *Pst* DC3000 in LB broth, containing the previously mentioned antibiotics, was pelleted twice at 3800 rpm for 15 min in 10 mM MgCl_2_ and resuspended in the starting culture volume (10 mL). The OD_600 nm_ of the suspension was read, then adjusted to get an inoculum of ~10^5^ CFU/mL for the challenge experiments while ~10^6^ CFU/mL was used for the gene expression experiments (see below Section 4.4). Twenty-four h after SID treatment, at least 6 leaves per plant were infiltrated with one mL of the prepared *Pst* DC3000 inoculum, using a 1-mL syringe [38]. At Day 0 and Day 3 post-infection, whole infected/infiltrated leaves were collected, weighed, and homogenized in microfuge tubes and plated on KB agar media containing rifampicin and kanamycin, to measure bacterial growth. Three infected leaves from one plant at Day 0, and 3 infected leaves from 3 plants at Day 3 were collected for bacterial counts for each treatment group. Experiments were independently repeated at least three times.

### 4.4. Characterization of Immunity Markers Induced by SID in Arabidopsis

To characterize molecular markers involved in the resistance induced by SID, five-week-old *Arabidopsis* Col-0 plants, grown in the same conditions as mentioned above, were treated with SID or 1% DMSO at the roots as described above. The leaves were infected 24 h later with ~10^6^ CFU/mL of *Pst* DC3000 or infiltrated with 10 mM MgCl_2_. In total, four groups were used: (i) a group treated with SID, then infected with *Pst* DC3000, (ii) a group treated with SID, and later infiltrated with 10 mM MgCl_2_ (mock infection), (iii) a group treated with 1% DMSO (the diluent for SID), then infected with *Pst* DC3000, and finally (iv) a group treated with 1% DMSO and infiltrated with 10 mM MgCl_2_. Eight h post-infection/infiltration, infected/infiltrated leaves were collected and immediately frozen in liquid nitrogen. 

Total RNA was extracted from leaves using the RNeasy Plant Mini Kit following the manufacturer’s (Qiagen, Germantown, MD, USA) recommendations. The extracted RNA was treated with DNase to remove all DNA residues, then quantified using a Nanodrop spectrophotometer (Nanodrop 2000, Thermo Scientific, Waltham, MA, USA). cDNA synthesis was performed with 2 µg total RNA using the SuperScriptII reverse transcriptase (Invitrogen, Burlington, ON, Canada). The target genes were quantified by qPCR using the Advanced qPCR Master Mix (Wisent, St-Bruno, QC, Canada) specific primers, and sterile water in a 20 µL total volume solution. The CFX96 system (Bio-Rad, Mississauga, ON, Canada) was used to detect the amplification level and was set with an initial step of 10 min at 95 °C followed by 40 cycles alternating between 15 s at 95 °C and 1 min at 60 °C. To analyze the relative expression of the target genes and immune defense pathways (Table 1), calculations were made using the ΔΔ cycle threshold (CT) mean from technical triplicates. Briefly, the threshold cycle (Ct) values of target genes were normalized to the endogenous control gene, the elongation factor 1 or *EF1* (ΔCT = Ct_target_ – Ct_endogenous_) and compared with a calibrator (ΔΔCT = ΔCt_sample_ – ΔCt_calibrator_). Relative expression was calculated using the following formula: RQ = 2^–ΔΔCT^. The specific primers for target genes used in this study were synthesized by The Integrated DNA Technologies (IDT Canada, Kanata, ON, Canada) and are listed in Table 1.

### 4.5. Statistical Analysis

GraphPad Prism version 8 (GraphPad, San Diego, CA; [39]) was used for all statistical analyses. Data were subjected to either *t*-test pairwise comparisons followed by Mann–Whitney post-test or a one-way analysis of variance (ANOVA), depending on the experiment. The ANOVA was used with Dunnett’s post-test to control for multiple comparisons. Specific tests are described in the ta legends.

## Figures and Tables

**Figure 1 pathogens-09-00806-f001:**
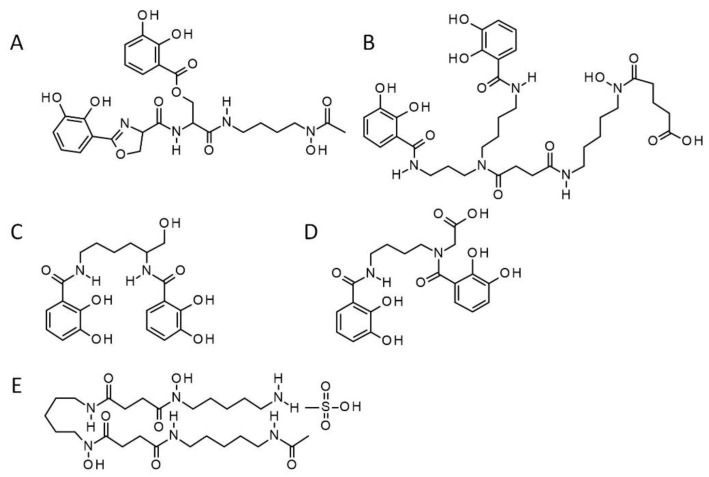
Chemical structures of natural and synthetic siderophores under study. The chemical structure of the natural mixed ligand siderophore fimsbactin produced by *Acinetobacter* spp. [20] shown in (**A**) can be compared to that shown in (**B**) for its synthetic analog, SID, a mixed ligand bis-catechol-mono-hydroxamate [22]. Similarly, in (**C**), myxochelin, a natural siderophore produced by *Azotobacter vinelandii* [23] can be compared to its mimetic (**D**), the synthetic bis-catechol, CAT [24]. Finally, the natural hydroxamate siderophore deferoxamine (DFO mesylate salt) is shown in (**E**). SID, CAT and DFO were used in this study. DFO is from Sigma-Aldrich (Oakville, ON, Canada).

**Figure 2 pathogens-09-00806-f002:**
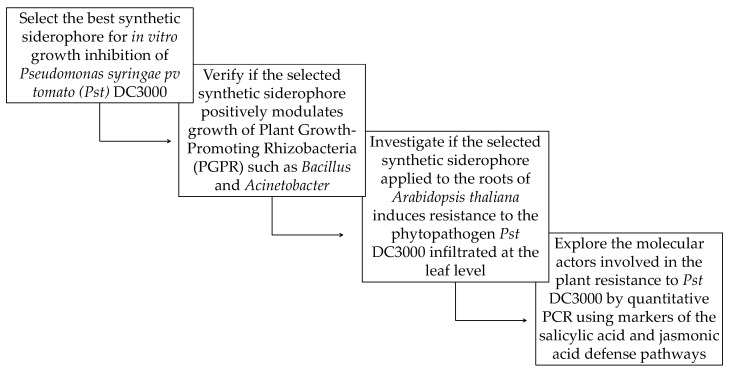
Flowchart summarizing the steps taken in this study to demonstrate the potential of synthetic siderophores, notably the mixed ligand bis-catechol-mono-hydroxamate siderophore (SID), as biocontrol agents against phytopathogens in agriculture.

**Figure 3 pathogens-09-00806-f003:**
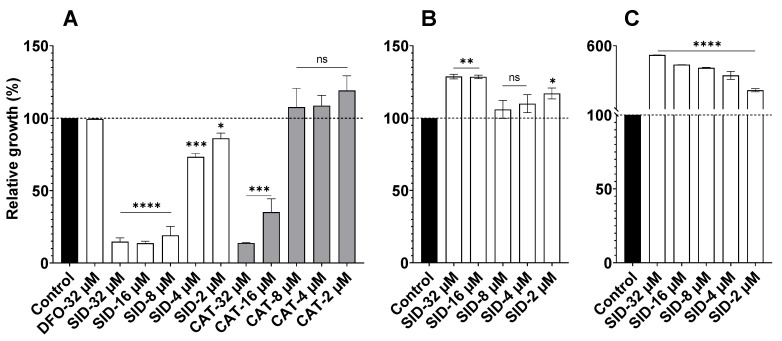
Effect of supplemental siderophores on growth of *Pst* DC3000, *B. subtilis* and of *A. baumannii.* Relative growth of *Pst* DC3000 (**A**), *B. subtilis* (**B**), and *A. baumannii* (**C**) in presence of synthetic siderophores (SID or CAT) was compared to the growth achieved (control, 100%) in the dipyridyl-containing iron-restricted medium in absence of any additional siderophore. The natural hydroxamate siderophore deferoxamine (DFO) was also used as comparator. Growth was measured by optical density (OD_595 nm_) after an incubation of 24 h. Data sets marked with asterisks are significantly different from the control as assessed by one-way ANOVA (*: *p* = 0.020; **: *p* = 0.001; ***: *p* = 0.0002; ****: *p* < 0.0001); ns: difference not statistically significant. Experiments were repeated at least three independent times.

**Figure 4 pathogens-09-00806-f004:**
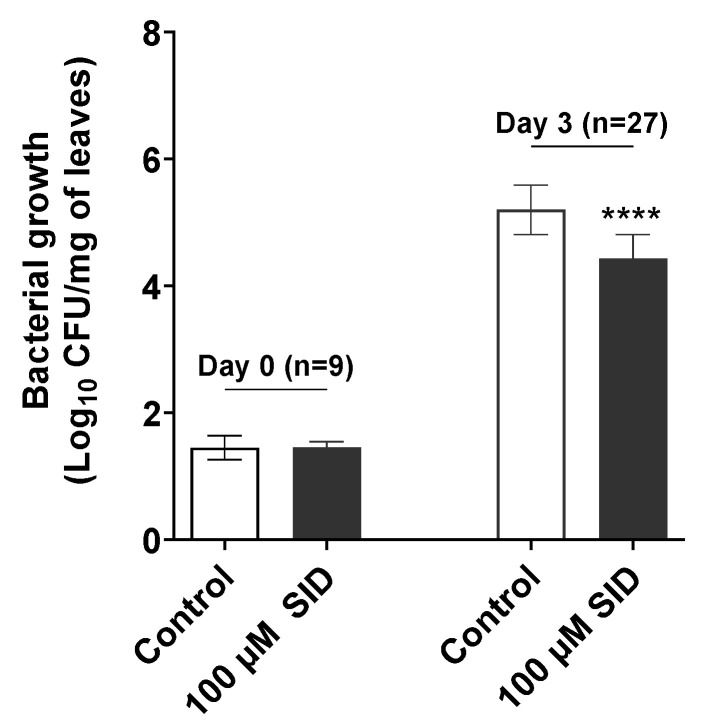
The synthetic SID induces systemic resistance of *Arabidopsis* Col-0 against *Pst* DC3000. Groups of plants roots were pre-treated with either a 10-mL of the SID solution at 100 µM or of the mock (control: 1% DMSO), then leaves were syringe infected with *Pst* DC3000 twenty-four h post treatment. Entire infected leaves were collected at the day of infection (Day 0) and three days after infection (Day 3) for bacterial counts. Data sets marked with asterisks are significantly different from the control as assessed by student’s test (****, *p* < 0.0001). The results were reproduced in three independent experiments.

**Figure 5 pathogens-09-00806-f005:**
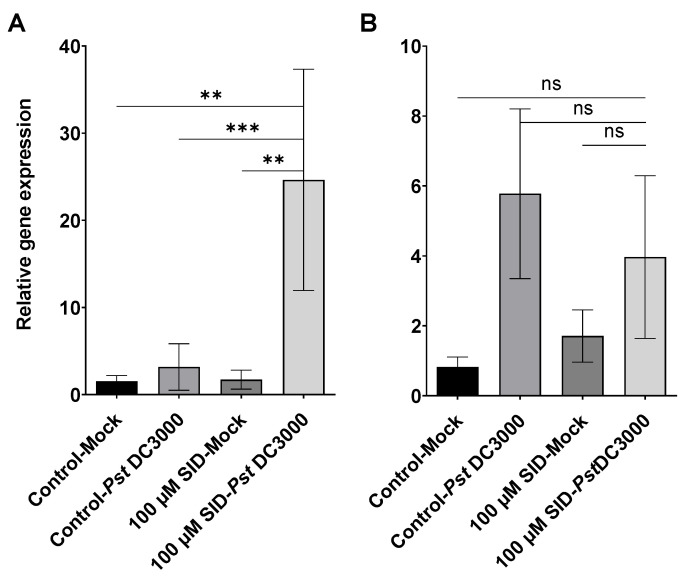
SID induces systemic priming of *PR1* (pathogenesis-related protein 1). Relative expression of the SA pathway marker, *PR1* (**A**) and the JA pathway marker, *PDF1.2* (**B**). The change in the *PR1* and *PDF1.2* mRNA levels are measured relative to those of the housekeeping gene *EF1* (Elongation Factor 1). Data sets marked with asterisks: *** (*p* < 0.0008); ** (*p* < 0.0015), are significantly different from the control as assessed by One-Way ANOVA; ns: difference is not significant. The control is the plants treated with the SID diluent (1% DMSO), and the mock infection is the diluent used in the bacterial inoculum (10 mM MgCl_2_). The results were reproduced in three independent experiments.

**Table 1 pathogens-09-00806-t001:** List of primers used in this study.

Gene	GenBank Accession	Pathway	Primer Sequence
*AtPR1*	AT2G14610	Salicylic acid	F: CTCATACACTCTGGTGGG
			R: TTGGCACATCCGAGTC
*AtPDF1.2*	AT5G44420	Jasmonic acid	F: TCTTTGGTGCTAAATCGTGTGT
			R: TGTAACAACAACGGGAAAATAAACA
*AtEF1*	AT5G60390	Elongation factor Tu	F: TCTCCGAGTACCCACCTTTGR: TCCTTCTTGTCCACGCTCTT

**Abbreviations**: *AtPR1*: *Arabidopsis thaliana* pathogenesis-related protein 1; *AtPDF1.2*: *Arabidopsis thaliana* plant defensin gene 1.2; *AtEF1*: *Arabidopsis thaliana* elongation factor 1.

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
