# Peer review of "A Siderophore Analog of Fimsbactin from Acinetobacter Hinders Growth of the Phytopathogen Pseudomonas syringae and Induces Systemic Priming of Immunity in Arabidopsis thaliana"

_pathogens, 2020, doi:10.3390/pathogens9100806_

Round 1
Reviewer 1 Report
This is a well written paper. The introduction gave enough background to the reader and the experiments were very well described. The conclusions are supported by the data.
Specific comments:
-Figure 1C showing the chemical structure of myxochelin seems unnecessary, as there is no reference of this figure in the text.
-Figure 2. The effect of DFO on Pst is not clearly explained. I assume that Pst has a transport system for DFO. If so, shouldn't it grow more than the control, which does not have any siderophore added? If Pst is using is own siderophore and adding another does not increase growth, why does Acinetobacter grow better when more SDI is added to the medium?. Please, explain.
-Figure 3. It is not clear why day 3 was used as the only time point besides day 0. It seems that more time points would give a better idea of the effect of SID on the bacterial load. Also, although statistically significant, how relevant is less than a log difference in bacterial load? Would that plant eventually be able to clear the infection?
-Figure 3. The choice of SDI dose (100uM) is also not explained. Have the authors tested if there is a dose-dependent effect?
-Figure 4. The authors show that SID triggers expression of resistance genes of the SA pathway but no of the JA pathway by looking at mRNA levels of 1 marker gene for each pathway. It would have made that data stronger if the authors had tested other genes of these two pathways. Also, there is no mention of the relevance of these findings.
Minor comments
-The scientific nomenclature for plants and bacteria should be formatted properly throughout the text
-A few characters are highlighted in yellow in the text with no obvious purpose.
Author Response
REVIEWER #1
This is a well written paper. The introduction gave enough background to the reader and the experiments were very well described. The conclusions are supported by the data.
Specific comments:
- Figure 1C showing the chemical structure of myxochelin seems unnecessary, as there is no reference of this figure in the text.
ANSWER: We revealed the structure of myxochelin in the figure to show it is the natural analog to the synthetic bis-catechol siderophore (CAT) that we used in the study (see Figure 2). This is similar to the synthetic mixed siderophore (SID) that we compare to its natural analog fimsbactin.
The reference to Figure 1 first appears in the Introduction section but reference to “Figure 1 and myxochelin” is in M&M section 4.1: “The chemical structures of fimsbactin and myxochelin A (Kunze et al. 1989), and those of their synthetic analogs SID and CAT, respectively, are presented in Fig. 1.”
- Figure 2. The effect of DFO on Pst is not clearly explained. I assume that Pst has a transport system for DFO. If so, shouldn't it grow more than the control, which does not have any siderophore added?
If Pst is using is own siderophore and adding another does not increase growth, why does Acinetobacter grow better when more SDI is added to the medium?. Please, explain.
ANSWER: Now Fig 3. The natural hydroxamate siderophore DFO was used as a control for its iron chelation ability while it is known that it does not promote or restrict Pst DC3000 growth in vitro at a concentration up to 1mM (Aznar et al., 2014). Figure 2A (now Fig. 3A) shows that DFO does not exacerbate the effect of iron deprivation in the media as it appears that Pst’s own siderophores (e.g., pyoverdine) can efficiently provide iron to bacteria in this iron-restricted environment. On the other hand, the synthetic bis-catechol (CAT) and the mixed ligands siderophore (SID) seem to sequester iron away from Pst DC3000 since they cause growth inhibition at concentrations higher than 2 mM for SID and higher than 16 mM for CAT.
The text was modified to clarify the results with DFO and Pst DC3000 in Fig. 2A (now Fig. 3A) (section 2.1, lines 117-128).
It has recently been shown that Acinetobacter species may or may not produce fimsbactin, but that all strains produce acinetobactin, which are chemically structurally close. Since those two siderophores are structurally related, they use the same uptake machinery (Bohac, Fang, Giblin, & Wencewicz, ACS Chem. Biol, 2019). Furthermore, these authors also showed that fimsbactin (or in our case, the synthetic SID) forms a 1:1 more stable complex with iron (III) than acinetobactin. It is thus expected that SID should have the upper hand in the complexing of iron (III) and be efficiently taken up by these bacteria, hence the growth promotion when SID is added in the medium.
The text was modified to clarify the results with SID and Acinetobacter in Fig. 2C (now Fig. 3C) (section 2.1, lines 135-140).
- Figure 3. It is not clear why day 3 was used as the only time point besides day 0. It seems that more time points would give a better idea of the effect of SID on the bacterial load. Also, although statistically significant, how relevant is less than a log difference in bacterial load? Would that plant eventually be able to clear the infection?
ANSWER: Now Fig. 4. The day-3 time point was used based on past experiences and experiments. Day 3 or 4 is the optimal time point that is very often used to see the effectiveness of Pseudomonas syringae pv tomato DC3000 infection and to pick up differences between treatment groups. See the following references by us and other groups: Laluk et al., The Plant Cell 2011, 23(8):2831-49 ; Gonzalez-Lamothe et al., The Plant Cell 2012, 24(2), 672–777; Huo et al., Nat Commun. 2017, 8(1):1808 ; Smakowska-Luzan et al., Nature. 2018, 553(7688):342-346 ; Peraki et al., Nature. 2018, 561(7722):248-252.
Plant specific immune responses to fungal and bacterial pathogens can be relatively moderate. The experimental infection we used (10e5 or 10e6 CFU directly infiltrated into the leaves) is quite challenging for the plant but using a lower inoculum leads to excessive experimental variations. Hence, in these conditions, a reproducible small reduction in bacterial load is significant. See the following references by us and other groups: Tsuda et al., PLoS Genet. 2009, 5(12):e1000772 ; Laluk et al., The Plant Cell 2011, 23(8):2831-49 ; Gonzalez-Lamothe et al., The Plant Cell 2012, 24(2), 672–777; Prokchorchik et al., New Phytol. 2020, 225(3):1327-1342).
- Figure 3. The choice of SDI dose (100uM) is also not explained. Have the authors tested if there is a dose-dependent effect?
ANSWER: Now Fig 4. Indeed, concentrations below 100 mM were initially used (1 to 16 mM) although with little beneficial effects observed. The use of small plant pots helped providing the necessary dose for obtaining differences between treatment groups. Unfortunately, we encountered problems when trying higher concentrations due to solubility issues. A sentence was added in the Results section 2.2 (line 179-180).
- Figure 4. The authors show that SID triggers expression of resistance genes of the SA pathway but no of the JA pathway by looking at mRNA levels of 1 marker gene for each pathway. It would have made that data stronger if the authors had tested other genes of these two pathways. Also, there is no mention of the relevance of these findings.
ANSWER: Now Fig 5. We believe the relevance of the finding was already mentioned in that the priming of resistance signal was sent from the roots to the rest of the plant using the salicylic acid pathway. The two markers used in this study are those typically used to give evidence on the activation of these two key pathways. See the following references as examples: Ward et al., The Plant Cell. 1991;3(10):1085–94; Koornneef et al., Commun. Integr. Biol. 2008, 1, 143–145; Leon-Reyes et al., Plant Physiol. 2009, 149, 1797–1809; Leon-Reyes et al., Planta 2010, 232 : 1423–1432; Aznar et al., Plant Physiology. 2014, 164(4):2167-83.
It would be indeed fascinating to explore more about these immune pathways. However, this would require the use of plant mutants and much more time investment at this stage. Future experiments will aim to further dissect the plant immune pathways involved in the systemic resistance mediated by this siderophore.
Minor comments
- The scientific nomenclature for plants and bacteria should be formatted properly throughout the text
ANSWER: It seems that all italicized words reverted to the normal font when the manuscript was uploaded on the journal platform. All plants and bacteria names are now italicized properly. The abbreviation of Pseudomonas syringae pv tomato DC3000 into Pst DC3000 is regularly used in the “plant world”. See for example Laluk et al., The Plant Cell 2011, 23(8):2831-49.
- A few characters are highlighted in yellow in the text with no obvious purpose.
ANSWER: Similarly, those special characters were “flagged” when the manuscript was uploaded on the journal platform. The yellow highlights were removed.
Reviewer 2 Report
In this paper, authors have taken four steps to investigate the role of a synthetic mixed ligand bis-catechol-mono-hydroxamate siderophore (SID). It mimics a natural siderophore's chemical structure, fimsbactin, produced by Acinetobacter spp. in the resistance against the phytopathogen Pseudomonas syringae pv tomato DC3000 (Pst DC3000), in Arabidopsis thaliana.
1- They have tested and confirmed the antibacterial activity of SID against Pst DC3000 in vitro.
2- They have tested and confirmed that the observed in vitro activity could translate into the resistance of Arabidopsis to Pst DC3000, using bacterial loads as endpoints in a plant infection model.
3- They have explored the molecular actors involved in the resistance of Arabidopsis induced by SID, using quantitative PCR,
4- Finally, they have tested in vitro the influence of SID on the growth of a reference PGPR, Bacillus subtilis, to assure that SID would not interfere with (plant growth-promoting rhizobacteria PGPRs)
The paper studies an interesting subject, and it is well written in English. The literature covers most of the related works, and there is a nice flow in the paper organization that makes it easier to understand.
However, I suggest some minor revises in the paper that might improve the article:
Some fonts are required to be changed: for example, all organisms should be in italic (e.g., Arabidopsis thaliana line 84), or some of the words are written in a different font/size (e.g., Line 131). In the abstract, PCR is used for the first time in an abbreviation version; the full version is expected for the first time.
In figure 1, the reference for the chemical structure is required. Also, in figure 2, mapping the relative growth to the relative growth rate ([0,1]) will make it simple for comparison.
Please cite the software GraphPad as follows: Swift, Mary L. "GraphPad prism, data analysis, and scientific graphing." Journal of chemical information and computer sciences 37.2 (1997): 411-412.
Since authors have taken many steps, a workflow/pipeline/flowchart that visualizes every action taken with the result allowed to take the next step will summarize the whole work and make it easier for the readers to follow the work.
Author Response
REVIEWER #2
Comments and Suggestions for Authors
In this paper, authors have taken four steps to investigate the role of a synthetic mixed ligand bis-catechol-mono-hydroxamate siderophore (SID). It mimics a natural siderophore's chemical structure, fimsbactin, produced by Acinetobacter spp. in the resistance against the phytopathogen Pseudomonas syringae pv tomato DC3000 (Pst DC3000), in Arabidopsis thaliana.
1- They have tested and confirmed the antibacterial activity of SID against Pst DC3000 in vitro.
2- They have tested and confirmed that the observed in vitro activity could translate into the resistance of Arabidopsis to Pst DC3000, using bacterial loads as endpoints in a plant infection model.
3- They have explored the molecular actors involved in the resistance of Arabidopsis induced by SID, using quantitative PCR,
4- Finally, they have tested in vitro the influence of SID on the growth of a reference PGPR, Bacillus subtilis, to assure that SID would not interfere with (plant growth-promoting rhizobacteria PGPRs)
The paper studies an interesting subject, and it is well written in English. The literature covers most of the related works, and there is a nice flow in the paper organization that makes it easier to understand.
However, I suggest some minor revises in the paper that might improve the article:
Some fonts are required to be changed: for example, all organisms should be in italic (e.g., Arabidopsis thaliana line 84), or some of the words are written in a different font/size (e.g., Line 131). In the abstract, PCR is used for the first time in an abbreviation version; the full version is expected for the first time.
ANSWER: Changes were made as requested.
In figure 1, the reference for the chemical structure is required.
ANSWER: The references were added in the Figure 1 legend.
Also, in figure 2 (now Fig. 3), mapping the relative growth to the relative growth rate ([0,1]) will make it simple for comparison.
ANSWER: We agree that the visualization of the relative growth can be improved. However, since growth is either increased or decreased, presenting the growth rate once the control is normalized to “1” could also be confusing. Instead, we have arranged graph 1C so that the 100% growth is at the same level as in A and B. We also added a dotted line at the 100% growth mark to facilitate comparisons.
Please cite the software GraphPad as follows: Swift, Mary L. "GraphPad prism, data analysis, and scientific graphing." Journal of chemical information and computer sciences 37.2 (1997): 411-412.
ANSWER: The reference is now cited in section 4.5
Since authors have taken many steps, a workflow/pipeline/flowchart that visualizes every action taken with the result allowed to take the next step will summarize the whole work and make it easier for the readers to follow the work.
ANSWER: A new Figure (Fig. 2) was added to show the flowchart and is referred to at the end of the Introduction section.